# Predictors of Adverse Pregnancy Outcomes in Pregnant Women Living with Obesity: A Systematic Review

**DOI:** 10.3390/ijerph19042063

**Published:** 2022-02-12

**Authors:** Romina Fakhraei, Kathryn Denize, Alexandre Simon, Ayni Sharif, Julia Zhu-Pawlowsky, Alysha L. J. Dingwall-Harvey, Brian Hutton, Misty Pratt, Becky Skidmore, Nadera Ahmadzai, Nicola Heslehurst, Louise Hayes, Angela C. Flynn, Maria P. Velez, Graeme Smith, Andrea Lanes, Natalie Rybak, Mark Walker, Laura Gaudet

**Affiliations:** 1Ottawa Hospital Research Institute, Ottawa, ON K1H 8L6, Canada; rfakhraei@ohri.ca (R.F.); kathryndenize@gmail.com (K.D.); alexandresimonmd@gmail.com (A.S.); aysharif@ohri.ca (A.S.); julzhu@toh.ca (J.Z.-P.); alyharvey@ohri.ca (A.L.J.D.-H.); bhutton@ohri.ca (B.H.); prattmisty@hotmail.com (M.P.); bskidmore@rogers.com (B.S.); springfall66@gmail.com (N.A.); alanes@bornontario.ca (A.L.); nrybak@ohri.ca (N.R.); mwalker@toh.ca (M.W.); 2School of Epidemiology and Public Health, University of Ottawa, Ottawa, ON K1G 5Z3, Canada; 3Institute of Health & Society, Newcastle University, Newcastle upon Tyne NE2 4AX, UK; nicola.heslehurst@newcastle.ac.uk (N.H.); louise.hayes@newcastle.ac.uk (L.H.); 4Department of Women and Children’s Health, King’s College London, London WC2R 2LS, UK; angela.flynn@kcl.ac.uk; 5Department of Obstetrics and Gynecology, Queen’s University, Kingston, ON K7L 3N6, Canada; maria.velez@queensu.ca (M.P.V.); graeme.smith@kingstonhsc.ca (G.S.); 6Department of Public Health Sciences, Queen’s University, Kingston, ON K7L 3N6, Canada; 7Department of Obstetrics and Gynecology, Faculty of Medicine, University of Ottawa, Ottawa, ON K1H 8M5, Canada; 8Department of Obstetrics, Gynecology and Newborn Care, The Ottawa Hospital, Ottawa, ON K1H 8L6, Canada; 9Department of Obstetrics and Gynecology, Kingston Health Sciences Centre, Kingston, ON K7L 2V7, Canada

**Keywords:** obesity, pregnancy, adverse outcomes, predictors

## Abstract

Obesity is a well-recognized risk factor for pregnancy complications. Most studies to date are in large cohorts, with results presented in a way that assumes all women living with obesity are at equal risk. This study investigates which women living with obesity are at higher risk of specific pregnancy complications. A systematic search of MEDLINE and Embase identified 7894 prospective or retrospective cohort studies exploring predictors of adverse outcomes among pregnant women living with obesity. Following screening, 61 studies were deemed eligible. Studies were selected if the effects of exposure to any predictor amongst pregnant women living with obesity could be collected. Maternal characteristics assessed for association with adverse outcomes included maternal age, race/ethnicity, maternal height, mode of conception, complement activation factors, and history of various comorbidities/procedures. Gestational diabetes mellitus was the most studied outcome (*n* = 32), followed by preterm birth (*n* = 29), preeclampsia (*n* = 27), low birthweight infants (*n* = 20), small for gestational age newborns (*n* = 12), and stillbirth (*n* = 7). This review identified important characteristics that should be considered during the screening and follow-up sessions of pregnant women living with obesity, including pre-existing type 1 diabetes, maternal age < 20 years or ≥35 years, non-White ethnicity, abdominal adiposity obesity, and history of bariatric surgery.

## 1. Introduction

Women living with obesity who become pregnant represent a population at risk of adverse outcomes for pregnancy and overall health issues, including diabetes and metabolic syndrome later in life [1,2]. With the increasing prevalence of obesity, the creation of screening and triaging tools to differentiate women at the highest risk of adverse outcomes from those at lower risk is essential to allow for the appropriate stratification of maternity care. Early identification of risk may lead to improved individual outcomes and decreased burden on health care systems. Many national obstetrics, pediatrics, and obesity organizations have identified maternal obesity as a critical health issue because it plays a direct role in both short- and long-term health outcomes for the mother and baby, including perpetuating the intergenerational cycle of obesity [3,4,5].

As of 2017, more than one in two adults and nearly one in six children in Organization for Economic Co-operation and Development (OECD) countries are living with overweight or obesity, with significant increases in rates over the last five years [6]. The increase in maternal weight has been accompanied by a concurrent increase in rates of pregnancy complications. Between 1996 and 2010 in Ontario, Canada, the rate of gestational diabetes mellitus (GDM) (2.7% to 5.6%) and pre-gestational diabetes (0.7% to 1.5%) doubled [7]. Rates of preeclampsia and gestational hypertension increased by 25% and 184%, respectively, in the United States (US) between 1987 and 2004 [8]. Although the gestational condition usually resolves during the postpartum, these women are at long-term risk of developing overt diabetes, chronic hypertension, and cardiovascular disease [9].

Given the costs of treating obesity and related sequelae, novel and innovative ways to identify women who should receive specialized prenatal care, while also avoiding unnecessary treatment for low-risk women living with obesity, are critical for improving outcomes. Currently, there is wide variation in the delivery of maternity care to women living with obesity, both in terms of the care provider (high-risk obstetrician, general obstetrician, family medicine and midwifery) and in locations where care is provided (tertiary referral center, community hospital, clinic or home) [10]. There is a lack of systematic reviews of primary research focused upon identifying risk factors that may place women living with obesity at an increased risk of adverse pregnancy outcomes. To address this knowledge gap, we performed a systematic review of published cohort studies exploring patient characteristics that may predict maternal and fetal outcomes in pregnant women living with obesity.

## 2. Materials and Methods

This review was registered in the PROSPERO database (CRD#42017060503). The protocol was developed by all members of the research team and was registered with the University of Ottawa Library’s online repository (https://ruor.uottawa.ca/handle/10393/35998 accessed on 1 March 2020). Deviations from the protocol are described in Appendix A that accompanies this review.

### 2.1. Electronic Literature Search

A search strategy was developed by an experienced information specialist (BS) with input from the research team. The search was conducted in January 2017 and was updated on 5 March 2020. Embase and MEDLINE electronic databases were searched. A combination of keywords and text terms were used. Literature searches were peer reviewed by a second independent information specialist using the established PRESS framework [11]. The search strategies used are provided in Appendix A to this review.

### 2.2. Study Eligibility Criteria

Eligibility criteria were established based on study population, measured exposures, clinical outcomes, and study design. Population studies that enrolled pregnant women meeting criteria for obesity (including class I obesity body mass index [BMI] 30.0–34.9 kg/m^2^, class II obesity BMI 35.0–39.9 kg/m^2^, and class III obesity BMI ≥ 40.0 kg/m^2^) were included, provided that the effects of the exposure within this population could be collected. For studies that were not exclusively conducted among women with obesity, only data relevant to women with obesity were extracted. Specific outcomes of interest included maternal mortality, stillbirths, preeclampsia, GDM, and low birth weight (LBW) including incidence of small-for-gestational-age (SGA) newborns. The outcomes of large-for-gestational-age (LGA) newborns and macrosomia were excluded from the analysis. Any identified potential exposure that may impact these outcomes of interest was included. We expected to identify information related to race/ethnicity, maternal age, and maternal comorbidities. In addition, we also expected to identify information regarding other risk factors not identified a priori. Non-randomized studies in the form of prospective and retrospective cohort studies were of interest; case-control studies were excluded. While no language restrictions were placed on the search, only studies published in English or French were retained. A detailed summary of eligibility criteria from all studies is provided in Appendix A.

### 2.3. Process of Study Selection

Duplicates from the bibliographic search were identified and removed. The remaining articles were uploaded into Distiller SR (Evidence Partners, Inc, Ottawa, ON, Canada) for level 1 (title/abstract) and level 2 (full text) screening. Both levels of screening consisted of two reviewers (two of MP, AS, and KD) screening for relevancy, first based on title and abstracts, and second based upon the full texts of the reports deemed potentially relevant. Conflicts were resolved by discussion. A Preferred Reporting Items for Systematic reviews and Meta-Analyses (PRISMA) flow diagram was completed to summarize the process of study selection [12].

### 2.4. Data Extraction Process

Data collection from the included studies was carried out by an expanded group of eight team members (with individual studies reviewed by two of MP, AS, LH, AF, NH, RF, JZ, ASh). A data extraction form was developed in Microsoft Excel and pilot tested by data collectors on a sample of studies. Each study was extracted by one reviewer and verified independently by a second reviewer. For all included studies, the following study characteristics were extracted: publication characteristics (authorship list; date of publication; journal of publication; and country/language of publication); study design information; outcomes assessed; demographics of the study population (including a priori eligibility criteria); focal predictor variables assessed for association (e.g., maternal age); statistical methods used to assess association with outcomes (e.g., univariate approaches such as contingency tables and multivariable approaches including logistic regression, proportional hazards modeling and linear regression); other risk factors, beyond the focal characteristics, included in the multivariable models performed if available (e.g., covariates adjusted for, to enhance comparability of findings across studies); and summary measures of association for the exposures evaluated, by outcome (e.g., odds ratios [OR], risk ratios with corresponding 95% confidence intervals). All data collected were compiled in structured tables to summarize key features of the included set of studies. Among the included studies that enrolled both women living with and without obesity, the effects of exposures were estimated for the target population using raw data or estimates of exposure effects when necessary.

### 2.5. Summarizing Study Findings

The primary objective of this review was the identification of patient characteristics associated with adverse maternal and fetal outcomes in pregnant women living with obesity. Given this objective, the identification/mapping phase of this work was considered a key part of this review. A narrative review of each outcome was written, with the intent to perform meta-analysis whenever possible. Forest plots and tables were constructed to provide Appendix A. The PRISMA 2020 Statement guided the reporting of the final review [12].

## 3. Results

### 3.1. Extent of Literature Found

A flow diagram documenting the process of study selection is presented in Figure 1. Following removal of duplicates, the literature search identified a total of 7894 unique titles and abstracts for review. Inspection of these citations by two reviewers excluded a total of 6692 citations due to not meeting inclusion criteria, leading to 1202 articles for review at full text. Amongst these citations, a total of 1141 were excluded by reviewers (reasons provided in Figure 1), leaving a total of 61 publications for final inclusion [13,14,15,16,17,18,19,20,21,22,23,24,25,26,27,28,29,30,31,32,33,34,35,36,37,38,39,40,41,42,43,44,45,46,47,48,49,50,51,52,53,54,55,56,57,58,59,60,61,62,63,64,65,66,67,68,69,70,71,72,73]. Table 1 provides an overview of the primary features of these studies.

### 3.2. Primary Study Characteristics

Included articles were published between 2009 and 2020, with the majority of articles published in 2019 (*n* = 19). Refer to Figure 2 for the number of articles published by publication year. The total number of pregnant women enrolled ranged between 50 [30] and 10,811,496 [60]. All studies involved populations of women living with obesity, although not all studies were conducted exclusively in that population (Table 1). Predictor and outcome data reported in the results of this review are only for women whose BMI was categorized as obese in the included studies. Variations in average maternal BMI and age were present across studies (see Appendix A).

The majority of studies included data for several thousand women (or births; Table 1), with total numbers of studies and corresponding percentage of studies evaluated provided in parentheses. A total of 35 studies (57.4%) [13,15,16,17,19,23,24,25,27,28,29,30,31,32,33,34,35,37,38,40,41,43,47,48,54,56,57,60,61,65,69,70,71,72,73] were conducted in the US, while others were conducted in Finland (*n* = 5; 8.2%) [21,22,49,55,67], Australia (*n* = 4; 6.6%) [14,36,45,46], France (*n* = 4; 6.6%) [26,52,58,68], Denmark (*n* = 2; 3.3%) [42,62], and one study (1.6%) from each of Canada [51], Egypt [18], England [63], Germany [39], Israel [66], Poland [59], Qatar [64], Saudi Arabia [53], Sweden [20], Singapore [44], and Turkey [50]. All studies were non-randomized; 13 (21.3%) used a prospective design [13,18,20,21,22,28,30,40,42,44,54,66,72], while 48 (78.7%) used a retrospective design [14,15,16,17,19,23,24,25,27,29,31,32,33,34,35,36,37,38,39,41,43,45,46,47,48,49,50,51,52,53,55,56,57,58,59,60,61,62,63,64,65,67,68,69,70,71,73].

### 3.3. Clinical Outcomes Evaluated

Table 1 presents a summary of the outcomes of interest that were reported by each of the included studies. Outcomes evaluated consisted of the following measures: GDM (32; 52.5%) [13,14,16,17,18,21,25,29,30,36,37,38,39,41,43,44,45,46,47,50,51,53,54,56,57,58,60,64,65,66,68,69]; LBW or SGA (30; 49.2%) [13,14,15,16,17,21,23,26,27,30,36,37,41,46,47,50,51,52,53,55,57,58,59,60,61,64,66,68,70,72]; preterm birth (29; 65.5%) [13,14,16,17,18,20,21,23,24,26,27,28,31,32,34,35,36,37,41]; preeclampsia (27; 44.3%) [13,15,16,17,18,19,20,21,22,28,30,33,37,40,41,42,43,44,46,50,51,52,53,58,62,64,71]; and stillbirths (7; 11.5%) [14,19,37,46,48,51,52]. There were no studies found that investigated maternal mortality among women living with obesity. Due to the absence of an accepted treatment for pregnancies suspected to expect LGA newborns or macrosomia, we did not include these outcomes in the present review.

### 3.4. Demographics Evaluated and Approaches to Statistical Analysis

Table 1 presents a listing of the focal risk factor(s) addressed within each study. Associations of maternal and fetal outcomes with a variety of patient characteristics were assessed amongst the included studies. The potential association of a priori outcomes with a broad range of risk factors was noted, with these covariates including race/ethnicity/indigenous status [14,15,17,25,29,35,36,38,39,64,69,70,71], history of bariatric surgery [19,26,41,44,47], history of sleep apnea [28,30], maternal age [21,23,32,33,37], mode of conception [16,66], pre-existing maternal conditions [20,22,48,49,52,55], parity [31,68], gestational weight gain (GWG) [32,43,56,57,58,59,60,65], maternal height [34], polycystic ovary syndrome (PCOS)/type of obesity [18,46], dietary inflammatory index (DII) [13], complement activation fragments [40], medication intake [42,45,53,67,74], laparoscopic sleeve gastrectomy [50], twin pregnancies [51], smoking status [54], interpregnancy maternal weight/BMI change [61,63], plasma adiponectin and leptin concentrations [62], and glucose challenge test [73]. Overall, 49 studies (77.0%) used multivariable logistic or linear regression analysis to account for the effects of confounding factors while exploring the association of a risk factor of focal interest with clinical endpoints [13,15,16,17,19,20,21,22,23,24,25,28,29,30,31,32,33,34,35,36,38,40,41,43,44,45,46,47,50,51,52,53,54,55,56,57,58,61,62,64,65,66,67,68,69,70,71,72,73]; the extent of adjustments was variable. Only unadjusted measures of association were available from 12 studies (23.0%) [14,18,26,27,37,39,42,48,49,59,60,63]. Estimates of measures of association for adjustment factors were not reported by most studies; however, these were not required to address primary study objectives. Meta-analyses were not performed because of the heterogeneity of the outcome assessment.

### 3.5. Risk Factors for Preeclampsia

Figure 3 present a summary of associations between the occurrence of preeclampsia and the risk factors assessed within the included studies. The collection of risk factors studied was diverse, and most estimates were derived from adjusted models; the number of variables adjusted varied (see Appendix A). Eight studies [21,33,37,44,50,51,52,62] assessed the impact of increased maternal age using different age categories; statistically significant differences suggesting a greater risk of preeclampsia with younger age (<20 years versus 20–24 years) was observed in one case [33], while another study suggested increased risk with greater age (>35 years versus <35 years) [21]. Data from four studies [15,17,64,71] were available to study the effects of race/ethnicity on preeclampsia risk, involving comparisons between White and Black women, Hispanic, Asian American, and Middle Eastern, American Indian/Alaskan Native women. One study observed significantly increased preeclampsia risk in Black and Asian American women living with class I or II obesity [17], another study found no significant difference in risk between Black or American Indian/Alaskan Native and White women nor between Middle Eastern and non-Middle Eastern women living with obesity [15,64,71]. Amongst other risk factors assessed, there was evidence of greater risk of preeclampsia in women with abdominal adiposity obesity than those with gynoid obesity; a history of type 1 diabetes mellitus [20]; a history of sleep apnea [28]; and no/late intake of multivitamins [42]. Effects of other risk factors included prior bariatric surgery/gastric bypass, complement activation factors, DII, vitamin D deficiency, and presence of PCOS; however, evidence of associations was not found.

### 3.6. Risk Factors for Low-Birth-Weight Newborns

Figure 4 present a summary of associations between the occurrence of LBW and SGA newborns and the risk factors assessed in relation to this outcome within the included studies. Most estimates were derived from adjusted models of varied complexity; there was also some variation in the definitions of both LBW and SGA newborns available across studies. Regarding maternal age, one study identified statistically significant increased risks of both LBW and very LBW newborns in women aged 40 years or older compared to women aged 20–29 years [37], whereas one study comparing women younger versus older than 35 years [21] of age and another study comparing births between adolescent and adult mothers [23] did not identify important differences between groups. Regarding race/ethnicity, Snowden et al. [17] identified statistically significant increased risk of LBW newborns in Black, Hispanic, and Asian American women living with obesity compared to White women living with obesity; Halloran et al. [27] and Marshall et al. [15] observed an analogous pattern, while Davies-Tuck et al. [14] found the risk of SGA newborns increased in South Asian versus Australian women. Amongst other risk factors assessed, there was evidence that greater risk of LBW or SGA in women with a history of bariatric surgery [41,50], laparoscopic sleeve gastrectomy [50], inadequate weight gain [57,58,60,61,66], and higher scores on the DII [13]. Additional risk factors explored were presence versus absence of sleep apnea [30], mode of conception [16], type of prior bariatric surgery [26], and GDM [52]; however, evidence of association was not found.

### 3.7. Risk Factors for Preterm Birth

A summary of associations between the occurrence of preterm birth and the risk factors assessed within the included studies is presented in Figure 5. The estimates of risk factors studied were predominantly derived from adjusted models. There was variation in the definitions of preterm birth available across studies, from <28 to <37 gestational weeks. Four studies [21,23,32,37] included analyses assessing the effects of increased maternal age using age groupings: one study observed increased risk of preterm births at <28 weeks [28,29,30,31] and 32–36 weeks in women aged ≥35 years versus <35 years [21]; the remaining three studies found no difference in risk [23,32,37]. Five studies [14,17,27,35,70] reported findings from analyses of race/ethnicity: Halloran et al. [27] and Anderson et al. [70] found no difference in risk among ethnicities, while Snowden et al. [17] noted increased risk of preterm birth <37 weeks in Black, Hispanic, and Asian American women compared to White women; Salihu et al. [35] also found increased risks in Black women compared to White women; while Davies-Tuck et al. [14] noted an increased risk in South Asian versus Australian women. Amongst other risk factors that were assessed, there was evidence of greater risk of preterm birth in women with abdominal adiposity obesity [18]; type 1 diabetes mellitus [20]; GDM [49]; nulliparity [31]; singleton [51]; vitamin D deficiency [53]; and lower GWG [24]. Additional risk factors explored included prior bariatric surgery/gastric bypass [26,41], history of sleep apnea [28], mode of conception [16], DII [13], and presence of PCOS; however, no evidence of associations was found.

### 3.8. Risk Factors for Gestational Diabetes Mellitus

Associations between the occurrence of GDM and the risk factors within the included studies are detailed in Figure 6. Given the substantial heterogeneity of diagnostic criteria for GDM, any studies with the diagnosis of GDM were included. Two studies looked at the effects of increased maternal age. One study found no difference in women aged ≥40 years versus 20–29 years [37]; however, another study reported statistically significance that GDM was higher among women who were 35 years or older [54]. Seven studies assessed the effects of race/ethnicity [14,17,39,53,54,64,69]. Snowden et al. [17] found that, compared to White women, Black women were at reduced risk of GDM, while risk was increased in Hispanic and Asian American women. Davies-Tuck et al. [14] reported more cases of GDM in South Asian versus Australian women, and Janevic et al. [69] reported that immigrants were at greater risk of GDM than US-born White women for all racial/ethnic groups, specifically for immigrant Indian women. Bar-Zeev et al. [54] reported that GDM was higher among non-Hispanic Black, Hispanic, and other racial-ethnic groups women compared to non-Hispanic White women. Similarly, Fallatah et al. [53] reported more cases of GDM in pregnant Saudi Arabian women than non-Saudi Arabian women, and Shaukat et al. [64] demonstrated a statistically significant difference in prevalence of GDM in Middle Eastern women compared to non-Middle Eastern women, while Reeske et al. [39] reported more cases of GDM in Turkish than German women. Amongst other risk factors that were studied, an increased risk of GDM was observed in women with abdominal adiposity obesity [18], women with BMI ≥ 50 kg/m^2^ [46], inadequate GWG [54,56,58,60,65], and in women living with class I obesity without a history of bariatric surgery (though this finding did not remain significant in sub-populations of class II or III obesity in the same study) [41]. Additionally, Karadag et al. [50] showed that laparoscopic sleeve gastrectomy may decrease the risk of GDM. Additional risk factors explored were the presence of sleep apnea, mode of conception, parity, and DII; however, evidence of a statistically significant association was not found.

Other studies involving women of a range of BMIs were also found. Kim et al. [25] conducted a series of multivariable regression analyses to evaluate the effects of ethnicity on the occurrence of GDM. In subgroups of women stratified by obesity class, risk ratios consistently showed increasing risk of GDM across BMI categories for all ethnicities, though risk increases were smaller in the Asian population than in White, African American, American Indian, and Hispanic populations [25]. In all cases, risks were increased in women within the same ethnicity as the class of obesity increased. A second study by Kim et al. [38] compared the prevalence of GDM between non-Hispanic Black and non-Hispanic White women, with an interest in whether the association varied by age; the study found that associations between risk of GDM and BMI did not vary across ethnicity or age.

Thrift et al. [36] used multivariable Poisson models to compare the incidence of GDM in Australian indigenous and non-indigenous women of varying pre-pregnancy BMIs. Separate analyses within the indigenous and non-indigenous groups were performed comparing women living with obesity and without obesity. Prevalence ratios were comparable in both the class I and II obese (indigenous: 3.00, 95% CI 2.47–3.63 versus non-indigenous: 2.76, 95% CI 2.64–2.88) and class III obese (indigenous: 4.44, 95% CI 3.48–5.67 versus non-indigenous: 4.47, 95% CI 4.18–4.78) categories [36].

Hedderson et al. [29] studied the effect of ethnicity and BMI on GDM rates in births in Northern California between 1995–2006 (*n* = 123,040 women of whom 40,279 had obesity). Separate multivariable logistic regression analyses (accounting for age at delivery, parity, gestational age, and educational attainment) were performed within each ethnic group, comparing the incidence of GDM amongst women with obesity to those with a recommended BMI. For women living with class I obesity, the magnitude of increased risk of GDM varied when compared to women living without obesity of the same ethnicity (non-Hispanic White: OR 3.71, 95% CI 3.26–4.22; African American: OR 4.08, 95% CI 2.94–5.66; Hispanic: OR 3.48, 95% CI 3.05–3.96; Asian: OR 2.45, 95|% CI 2.05–2.94; Filipina: 2.87, 95% CI 2.32–3.55).

### 3.9. Risk Factors for Stillbirth

Few studies reported findings related to the occurrence of stillbirth, and all assessed associations involving different risk factors (Table 1). In a retrospective cohort study, Davies-Tuck et al. [14] looked at singleton births from South Asian born women (*n* = 875) and Australian/New Zealand born women (*n* = 5163) and for the occurrence of stillbirths, an unadjusted OR of 0.88 (95% CI 0.36–2.10) comparing women of Australian/New Zealand descent to the reference group of South Asian women found insufficient evidence of a difference between groups.

Parker et al. [19] studied an association between prior bariatric surgery and adverse outcomes in pregnancy in a retrospective cohort study of 186,605 women living with obesity with singleton pregnancies. In comparing the incidence of stillbirths in women with a history of the procedure (*n* = 1585) to the control group of women who did not (*n* = 185,120), a multivariable regression model accounting for several covariates (including age, race, pre-existing diabetes, GDM, smoking, and hypertension) found no evidence of a difference between groups (adjusted OR 0.83, 95% CI 0.13–5.36).

Barton et al. [37] reported findings from a retrospective cohort study that assessed pregnancy outcomes of interest in women ≥ 40 years of age; analyses were undertaken in populations of both women living with (*n* = 9452) and without (*n* = 44,028) obesity. In the group of women living with obesity, stillbirths were observed more commonly in the group of 228 women aged ≥ 40 years (0.9%) compared to the group of 9224 women aged between 20–29 years (0.3%); however, an unadjusted OR comparing groups was associated with considerable uncertainty (OR 2.91, 95% CI 0.69–12.27).

Pratt et al. [46] reported findings from a retrospective cohort study based in Australia that assessed pregnancy outcomes of interest in women with BMI ≥ 50 kg/m^2^. The study assessed comparisons among 18,518 women with singleton pregnancies separated in categories of BMI; it was reported that nine stillbirths occurred ≥40 gestational weeks. Stillbirths ≥40 weeks made up 4% of stillbirths in women living with obesity (*n* = 2/50); however, there were no stillbirths in women with BMI ≥ 50 kg/m^2^. The study reported there was no increased risk of stillbirth with increasing maternal BMI; however, as mentioned by the investigators, stillbirth is an uncommon outcome, and their sample size was not powered to detect a significant difference in stillbirth by maternal BMI [46].

Browne et al. [48] reported findings from a retrospective cohort study that assessed the joint effects of obesity and pre-gestational diabetes on the risk of stillbirth in pregnant women. The study assessed comparisons among 3,097,123 women with singleton non-anomalous births in each BMI class that was stratified into the four gestational age periods for analysis: 24–33, 34–36, 37–39, and 40–42 weeks. The overall rate of stillbirth increased 78% in pregnancies of women who had class III obesity compared to women in the recommended BMI category. The rate of stillbirth further increased with coexistence of pre-gestational diabetes in the class III obesity group. The study reported the highest risk of stillbirth between the gestational age of 37 and 39 weeks; when the adjusted hazard ratio in the diabetic recommended BMI group was 9.63 (95% CI 5.65–16.40), the adjusted hazard ratio in the diabetic class III obesity group was 25.34 (95% CI 15.58–41.22) [48].

## 4. Discussion

This systematic review has identified specific predictors of adverse maternal and newborn outcomes among women with obesity. Predictors such as prepregnancy type 1 diabetes non-White ethnicity, specific groups of maternal age (<20 years and ≥35 years), abdominal adiposity obesity, and history of bariatric surgery were found to increase the risk of various outcomes such as preeclampsia, LBW/SGA, preterm birth, GDM, and stillbirth.

The outcomes for this study were chosen because they represent conditions for which screening is appropriate and potentially beneficial (i.e., prediction can allow directed surveillance or treatment that reduces risk). Other adverse outcomes exist that were not included in our review included LGA newborns and maternal mortality. Large-for-gestational-age newborns were clearly demonstrated to be more common in pregnancies of women who live with obesity and result in increased risks of caesarean section, birth injury, and admission to the neonatal intensive care unit. However, the ability to accurately predict and treat LGA prenatally remains limited–late pregnancy ultrasounds generally have a ±10% range of error and even when accurate, it is not yet possible to predict which LGA newborns will encounter birth complications. The study of rare outcomes, such as maternal mortality, remains very challenging. Even in extremely large cohorts, maternal mortality is usually not captured effectively as it is virtually impossible to guarantee privacy of information for such rare occurrences. The study of maternal mortality is currently limited to national venues such as the MBRRACE-UK Confidential Enquiry into Maternal Death conducted by the National Health Services in the United Kingdom [74].

Data from our study support the intuitive concept that degree of maternal obesity and degree of maternal risk are positively correlated. Individual levels of health vary at a given BMI, a concept that must be considered and respected. Nevertheless, within the healthcare system, both physical equipment and staffing resources needed to provide optimal maternity care to patients with the highest BMIs are most available at centres with the highest levels of care. Consideration should be given to early referral to higher-level care centres when directing referrals for individual patient care.

Gestational weight gain has been shown to impact many of the adverse pregnancy outcomes included in this review and is an important modifiable risk factor. The purpose of this review was to supplement clinical decision-making at the initial assessment for prenatal care regarding optimal care environment (e.g., maternity care provider, location of prenatal care provision). For this reason, GWG was beyond the scope of the current review.

The means by which maternal obesity impacts health of mother and child (beginning in utero and extending throughout life) are being sought. There is increasing evidence to implicate obesity-associated low-level inflammatory mediators as important contributors. This model links maternal obesity and/or an obesogenic diet with altered adipokine secretion, insulin resistance, and increased circulating lipids, which in turn are associated with increased levels of markers of inflammation (including interleukin-, interleukin-1β and tumor necrosis factor-α). Elevated levels of these factors have been associated with placental inflammation, altered placental nutrient transport, and altered placental structure, all of which have the potential to negatively affect developmental programming of fetal metabolism and increase lifelong risks of obesity and metabolic syndrome.

This extensive systematic review highlights the vast amount of data available from studies of the outcomes of pregnancy in women living with obesity. Dozens of studies have been reported, the majority with similar findings. While they clearly demonstrate increased risks of common complications of pregnancy at population levels, the included studies do not allow the individual maternity care provider or the patient themselves any insight into their individual risk. Despite a plethora of available data, we were unable to find characteristics other than perhaps age and ethnicity that could reliably identify women living with obesity who are at high risk of adverse pregnancy outcomes–nor could meta-analysis be performed due to inherent heterogeneity. Personalized medicine (the provision of care that is tailored to the individual patient) has the potential to optimize outcomes at the individual level and is now widely utilized in many areas of medicine.

There are currently two studies available that address the prediction of uncomplicated pregnancies in women living with obesity [75,76]. The UPBEAT Consortium in the United Kingdom published a prediction model in 2017 for uncomplicated pregnancies in women with obesity that was developed using a prospective multicentre cohort [75]. In their study, 505/1409 (36%) of women living with class I obesity had uncomplicated pregnancy and birth. The significant predictors of uncomplicated pregnancy presented were multiparity, increased plasma adiponectin, maternal age, systolic blood pressure, and hemoglobin A1C. When only clinical factors were included in the model, the team was able to achieve sensitivity, specificity, positive, and negative predictive values of 31%, 86%, 56%, and 69%, respectively. Although the UPBEAT model sought to predict uncomplicated rather than complicated pregnancies, several common predictors emerged. The clinical features that correlated with uncomplicated pregnancy outcomes are, not surprisingly, the opposite of the features that we identified as predictors of complicated pregnancies.

In a large population cohort in Ontario Canada [76], our team has demonstrated that more than half of women with obesity who have no other pre-existing medical or early obstetric complicating factors proceed through pregnancy without adverse obstetric complication. Of the studied Ontario maternity population, 17.7% (*n*  =  117,236) women were living with obesity. Of these 20.6% had pre-existing co-morbidities or early obstetric complicating factors. Amongst women living with obesity but without early complicating factors, 58.2% (*n*  =  54,191) experienced pregnancy without complication; this is in comparison to 72.7% of women living with a healthy weight and with no early complicating factors. Women living with obesity and no early pregnancy complicating factors are more likely to have an uncomplicated pregnancy if they are multiparous, younger, more affluent, of White or Black ethnicity, of lower weight, with normal placental-associated plasma protein-A, and/or spontaneously conceived pregnancies.

The data we have generated regarding predictors of both complicated and uncomplicated pregnancies in women living with obesity will allow for the development of tools to assist maternity care providers in providing more precise risk estimates and care plans for patients. For example, if there is a pre-existing diagnosis of diabetes, whether treated with oral medications or insulin, or if the hemoglobin A1C in the first trimester of pregnancy is ≥5.7%, the risk of adverse pregnancy outcomes is significantly increased and additional care during pregnancy and birth is recommended. Similarly, either a pre-existing diagnosis of hypertension (especially if anti-hypertensive treatment is used) or a mean arterial pressure of ≥90 mmHG detected in the first trimester correlates strongly with adverse placenta-mediated pregnancy complications. Such patients benefit from increased maternal and fetal surveillance, preventative treatments such as aspirin (~150 mg po qhs from 12–35 weeks), and consideration of timing of delivery. When alterations in prenatal care are appropriate, screening for at-risk pregnancies would allow for triaging of prenatal care to the optimal site and care provider and promote more tailored treatment for individual pregnancies.

In addition to the findings of our review, there are some situations in which tertiary level care is recommended due to the increased risk of pregnancy complications. Women with a history of bariatric surgery (particularly malabsorptive procedures), regardless of their current weight, benefit from care in a multidisciplinary team that includes Maternal Fetal Medicine due to increased risks of fetal growth restriction, preterm birth, and perinatal death. Women living with obesity and a multiple gestation should also be cared for in a tertiary setting [77,78]. Finally, women living with the highest levels of obesity have significant risk of complications related to anesthesia, including difficulty with or inability to place a regional technique (including epidural and spinal), difficult airway due to excess neck adiposity, and difficulty with ventilation due to chest wall adiposity/weight [77]. For these reasons, delivery in a centre with access to specialized anesthesia support should be considered for women living with a pre-pregnancy BMI 40 kg/m^2^ or greater.

There are limitations of the current review to be noted. This review represents a synthesis of both prospective and retrospective cohort studies of highly variable size, some of which may be more prone to the impact of bias than others. Most studies reported findings for a focal patient characteristic of interest from a multivariable statistical model that accounted for the potential effects of other confounding factors; however, the extent of adjustment varied across studies. Many of the studies enrolled a combination of women with varying BMI levels, including women with a recommended BMI, and thus in many cases, demographic information for the subgroup of women living with obesity was not available; this limited our ability to assess the similarity of women across studies. The extent of obesity, in terms of severity relating to obesity class, is likely to have varied across studies and thus may influence findings to some degree. In a smaller number of cases, we encountered studies that enrolled a mixture of women living with and without obesity wherein the approach to multivariable modeling and reporting did not inform an understanding of the effects of focal covariates in the population living with obesity. In such cases where possible, we reconstructed the numbers of events in the different subgroups of interest within the population living with obesity and calculated unadjusted OR. Given the high degree of variation in outcomes, exposures, study designs, and approaches to data analysis, we prepared a clear overview of the findings observed across a wide body of literature. Finally, we did not attempt to categorize preterm birth into spontaneous and iatrogenic, a distinction that also impacts the outcome of low birthweight. Prematurity and low birthweight are both highly related to underlying pregnancy complications, such as preeclampsia.

While most of the research to date uses BMI to define maternal obesity, BMI has some limitations (including its categorical nature, the lack of differentiation between weight of muscle and weight of fat, and the lack of gender- and pregnancy-specificity, among others), and the best method of screening for adiposity in pregnancy has yet to be determined. Studies to date have explored associations between adverse pregnancy outcomes of interest in this systematic review with early, mid-, and late pregnancy anthropometric measures such as waist circumference, waist hip ratio, and mid-upper arm circumference [79,80], bio-electrical impedance measures of body fat percentage, fat mass and fat-free mass [81,82], and ultrasound scan measures of subcutaneous and visceral fat thickness [83,84]. In the future, a more optimal clinical measure of maternal adiposity than BMI may be identified and could be used in prediction modelling.

This review represents the launching point for a number of future research endeavors. Firstly, we are developing a clinical triage tool (calculator) that will use a few easily available data points (such as age, BMI, presence of pre-existing diabetes, presence of pre-existing hypertension, etc.) to present an individualized risk of pregnancy complications. Maternity care providers and patients can then consider how, where, and from whom the patient should receive their prenatal care. Finally, further investigation to determine the ability of urine protein levels, first trimester serum analytes (including pregnancy-associated plasma protein A and placental growth factor), nutritional markers (e.g., hemoglobin, ferritin, B_12_, vitamin D), and ultrasound markers (for example, uterine artery Doppler resistance) in predicting adverse pregnancy outcomes in women living with obesity would be beneficial.

## 5. Conclusions

We have demonstrated that there is ample existing research on the predictors of adverse pregnancy outcomes in women living with obesity. Further, we have clearly identified important predictors of those adverse outcomes, namely, pre-existing type 1 diabetes, maternal age <20 years or ≥35 years, non-White ethnicity, abdominal adiposity obesity, and history of bariatric surgery. Maternity care providers should have heightened awareness of the increased risk of preeclampsia, GDM, LBW/SGA, preterm birth, and stillbirth in these pregnancies and consider the implications for maternity care provision locally and regionally. We are using the information from this extensive review to develop clinical tools that will assist providers in personalizing care to patients and allow more appropriate resource allocation. Personalizing maternity care provision will result in the best patient-centred care at the best time in the best place for pregnancies in women living with obesity.

## Figures and Tables

**Figure 1 ijerph-19-02063-f001:**
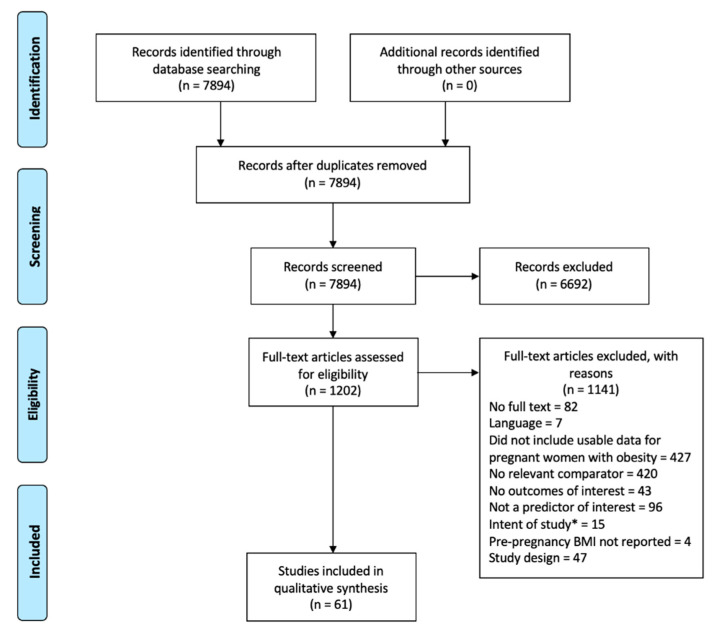
PRISMA. * Intent of study indicates that the objective of the study was not to study the predictors of adverse pregnancy outcomes.

**Figure 2 ijerph-19-02063-f002:**
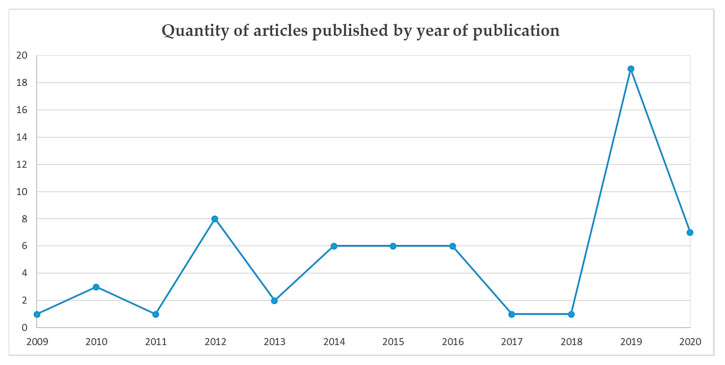
Graph illustrating the number of articles published over time.

**Figure 3 ijerph-19-02063-f003:**
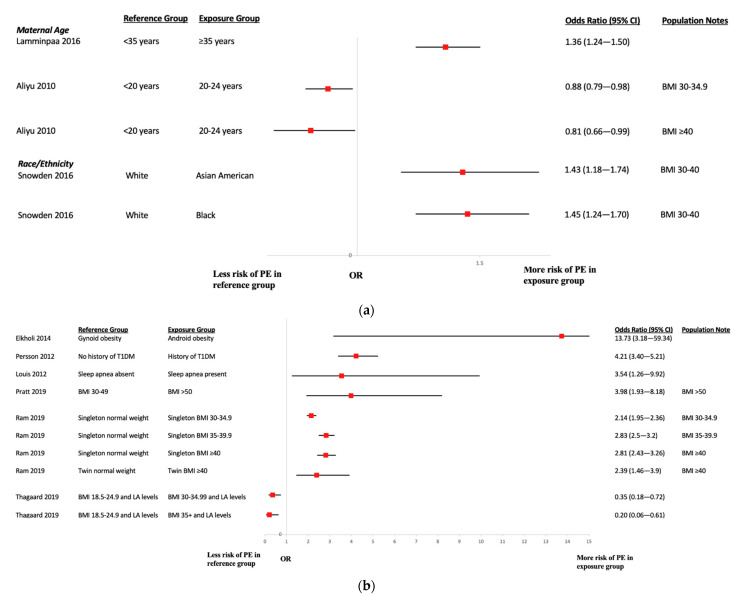
(**a**) PE Age and Race. Acronyms: BMI = body mass index; CI = confidence interval; PE = preeclampsia. (**b**) PE Risk Factors. Acronyms: BMI = body mass index; T1DM = type 1 diabetes mellitus; LA= log adiponectin; CI = confidence interval; PE = preeclampsia.

**Figure 4 ijerph-19-02063-f004:**
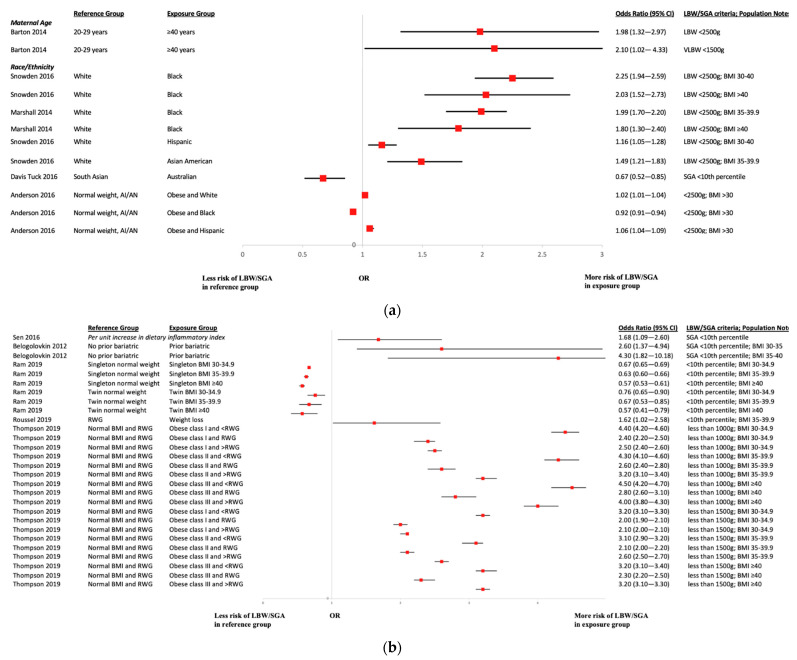
(**a**) LBW Age and Race. Acronyms: BMI = body mass index; LBW = low birth weight; VLBW = very low birth weight; AI/AN = American Indian/Alaskan Native; SGA = small-for-gestational-age; CI = confidence interval. (**b**) LBW Risk Factors. Acronyms: BMI = body mass index; LBW = low birth weight; VLBW = very low birth weight; RWG = recommended weight gain; SGA = small-for-gestational-age; CI = confidence interval.

**Figure 5 ijerph-19-02063-f005:**
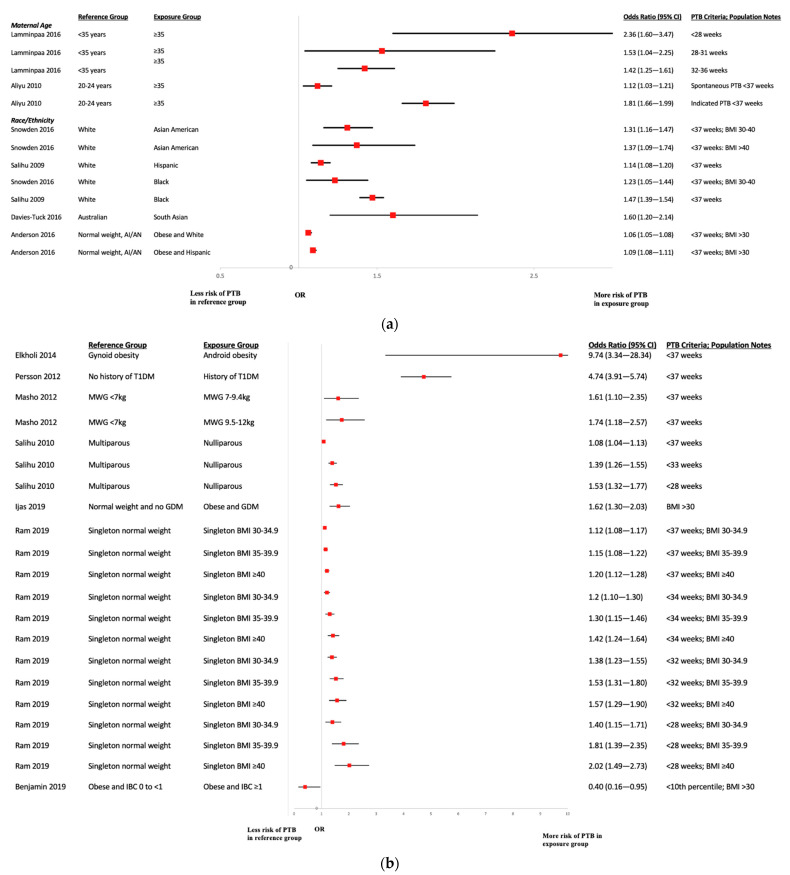
(**a**) Pre-term Birth Age and Race. Acronyms: BMI = body mass index; PTB = preterm birth; AI/AN = American Indian/Alaskan Native; CI = confidence interval. (**b**) Pre-term Birth Risk Factors. Acronyms: BMI = body mass index; PTB = preterm birth; T1DM = type 1 diabetes mellitus; MWG = maternal weight gain; GDM = gestational diabetes mellitus; IBC = interpregnancy-BMI change; CI = confidence interval.

**Figure 6 ijerph-19-02063-f006:**
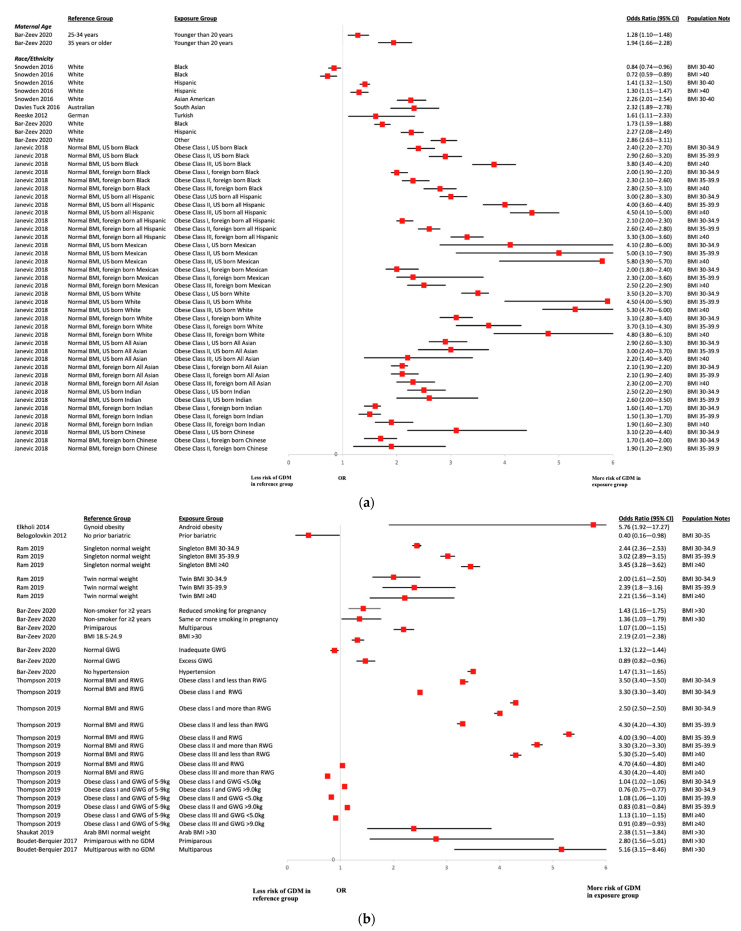
(**a**) GDM Age and Race. Acronyms: BMI = body mass index; US = United States; CI = confidence interval. (**b**) GDM Risk Factors. Acronyms: BMI = body mass index; RWG = recommended weight gain; GWG = gestational weight gain; GDM = gestational diabetes mellitus; CI = confidence interval.

**Table 1 ijerph-19-02063-t001:** Overview of Included Studies.

Author (Year)	Countries of Conduct	Setting	Total Study Size	Focal Risk Factor Evaluated	Outcomes Reported				
					Preeclampsia	Low Birth Weight (and SGA)	Gestational Diabetes	Preterm Birth	Stillbirth
Sen [13] (2016)	USA	PC	261 *	DII (change per unit increase)	X	X	X	X	
Davies-Tuck [14] (2016)	Australia	RC	6038 *	Race (AUS/NZ versus South Asian)		X	X	X	X
Marshall [15] (2014)	USA	RC	61,191 *	Race (Caucasian versus African American)	X	X			
Machtinger [16] (2015)	USA	RC	1015 *	Mode of conception (spontaneous versus IVF)	X	X	X	X	
Snowden [17] (2016)	USA	RC	76,174 *	Race (Caucasian, Hispanic, African American, Asian American)	X	X	X	X	
Elkholi [18] (2014)	Egypt	PC	400	PCOS (yes versus no), Obesity type (android versus gynoid)	X		X	X	
Parker [19] (2015)	USA	RC	186,705	History of gastric bypass surgery (yes versus no)	X				X
Persson [20] (2012)	Sweden	PC	82,949 *	History of type 1 diabetes (yes versus no)	X			X	
Lamminpaa [21] (2016)	Finland	PC	29,995 *	Advanced maternal age (≥35 y versus >35 y)	X	X	X	X	
Metsälä [22] (2015)	Finland	PC	11,404 *	Histories of diabetes, hypertension (yes versus no)	X				
Houde [23] (2015)	USA	RC	790,721 *	Maternal adolescent age (12–19 versus ≥20); adequacy of GWG		X		X	
Masho [24] (2012)	USA	RC	2960 *	Maternal weight gain during pregnancy(quartiles)				X	
Kim [25] (2013)	USA	RC	251,237 *	Ethnicity (white, black, Asian, Hispanic, American Indian)			X		
Ducarme [26] (2013)	France	RC	79 women (94 pregnancies)	Type of bariatric surgery (LAGB versus RYGB)		X		X	
Halloran [27] (2012)	USA	RC	2815 *	Ethnicity (Caucasian versus African American)		X		X	
Louis [28] (2012)	USA	PC	161	Obstructive sleep apnea (yes versus no)	X			X	
Hedderson [29] (2012)	USA	RC	40,279 *	Race (White, Hispanic, African American, Asian, Filipina)			X		
Olivarez [30] (2011)	USA	PC	50 *	Obstructive sleep apnea (yes versus no)	X	X	X		
Salihu [31] (2010)	USA	RC	132,894 *	Nulliparity (nulliparious versus multiparous), race (white, black, hispanic)				X	
Aliyu [32] (2010)	USA	RC	3278 *	Maternal age (20–24 versus ≥35)				X	
Aliyu [33] (2010)	USA	RC	51,427 *	Adolescent maternal age (<20 years versus 20–24 years)	X				
Shachar [34] (2015)	USA	RC	19,664 *	Maternal height (five size categories considered)				X	
Salihu [35] (2009)	USA	RC	149,532 *	Ethnicity (Hispanic, Caucasian, African American)				X	
Thrift [36] (2014)	Australia	RC	55,275 *	Indigenous status (indigenous versus non-indigenous)		X	X	X	
Barton [37] (2014)	USA	RC	9452 *	Maternal age (20–29 versus >40)	X	X	X	X	X
Kim [38] (2014)	USA	RC	462,296 *	Ethnicity (Caucasian, African American), Age (20–29, >40)			X		
Reeske [39] (2012)	Germany	RC	3338	Ethnicity (Turkish versus German)			X		
Lynch [40] (2012)	USA	PC	1013 *	Complement activation fragments (Bb, C3a; quartiles)	X				
Belogolovkin [41] (2012)	USA	RC	131,166 *	Prior bariatric surgery (yes versus no)	X	X	X	X	
Hogh [42] (2020)	Denmark	PC	15,154 *	Multivitamin use (non-users versus periconceptional use versus early pregnancy use)	X				
Njagu [43] (2020)	USA	RC	374 *	GWG (≤20 lbs versus >20 lbs weight gain)	X		X		
Malik [44] (2020)	Singapore	PC	55 *	Postbariatric surgery (yes versus no)	X		X		
Porteous [45] (2020)	Australia	RC	5426	Referral to an Ante-natal dietitian (yes versus no); number of appointments attended			X		
Pratt [46] (2019)	Australia	RC	18,402 *	Hypertensive disorder	X	X	X		X
Dolin [47] (2019)	USA	RC	76 *	Bariatric surgery (<12 months before versus ≥12 months before)		X	X	X	
Browne [48] (2019)	USA	RC	3,097,123 *	Diabetes (nondiabetic versus pregestational diabetic)					X
Ijas [49] (2019)	Finland	RC	24,577 *	Age (<19, 20–29, 30–39, ≥40), parity (primiparous versus multiparous), SES (upper, lower, manual, other)				X	
Karadag [50] (2020)	Turkey	RC	144 *	LSG (≤1 year versus >1 year before pregnancy)	X	X	X	X	
Ram [51] (2019)	Canada	RC	487,870 *	Singleton versus twin pregnancies	X	X	X	X	X
Meghelli [52] (2020)	France	RC	472 *	Age, GWG, hospitalization	X	X		X	X
Fallatah [53] (2019)	Saudi Arabia	RC	132 *	Vitamin D levels (deficient versus optimal versus therapeutic versus excess)	X	X	X	X	
Bar-Zeev [54] (2020)	USA	PC	222,408 *	Prenatal smoking (non-smoker, quit smoking, reduced the amount smoked, smoked the same or more)			X		
Kong [55] (2019)	Finland	RC	649,043 *	Prematurity, diabetes (no versus insulin treated versus type II)		X			
Ukah [56] (2019)	USA	RC	165,908 *	GWG; race (Black, Native American, Hispanic, White)			X	X	
Feghali [57] (2019)	USA	RC	5814 *	GWG (adequate, inadequate, excess)		X	X	X	
Roussel [58] (2019)	France	RC	996 *	GWG (recommended weight gain, low weight gain, weight loss)	X	X	X		
Nowak [59] (2019)	Poland	63RC	474 *	GWG (inadequate versus excess)		X			
Thompson [60] (2019)	USA	RC	10,811,496 *	GWG (<5 kg, 6–9 kg, >9 kg), gestational hypertension		X	X	X	
Benjamin [61] (2019)	USA	RC	694 *	LGA, GWG (inadequate versus adequate versus excess), height (<1.60 m, 1.6 m to <1.65 m, 1.65 m to <1.7 m, ≥1.7 m)		X		X	
Thagaard [62] (2019)	Denmark	RC	2503 *	Adiponectin and leptin concentrations	X				
Grove [63] (2019)	England	RC	20,069 *	GWG (decrease in BMI versus increase in BMI)				X	
Shaukat [64] (2019)	Qatar	RC	1134 *	Ethnicity (Arab versus non-Arab), hypertension	X	X	X	X	
Moore Simas [65] (2019)	USA	RC	2039 *	GWG (low, appropriate, excess), AGT (yes versus no)			X		
Frankenthal [66] (2019)	Israel	PC	1058 *	GWG (low, appropriate, excess), assisted reproduction treatment (yes versus no)		X	X		
Laine [67] (2019)	Finland	RC	6920 *	Antidepressant use (yes versus no)				X	
Boudet-Berquier [68] (2017)	France	RC	3208 *	Parity (primiparous versus multiparous), GWG (low, appropriate, excess), hypertensive complications (yes versus no), vaginal birth (yes versus no), smoking (non-smoker, quit smoking, smoke the same); maternal age (18–24, 25–29, 30–34, ≥35)		X	X		
Janevic [69] (2018)	USA	RC	668,035 *	Ethnicity (Black, White, all Hispanic, all Asian, Mexican, Chinese, Indian); place of birth (foreign born versus USA born)			X		
Anderson [70] (2016)	USA	RC	5,193,386 *	Ethnicity (American Indian/Alaska Native, Black, White, Hispanic)		X		X	
Zamora-Kapoor [71] (2016)	USA	RC	71,080 *	Ethnicity (American Indian/Alaska Native versus White)	X				
Gernand [72] (2014)	USA	PC	792 *	Vitamin D status		X			
Subramaniam [73] (2015)	USA	RC	14,525 *	LGA, macrosomia, shoulder dystocia, hypertension	X				

Note. This table presents a comprehensive summary of the outcomes of interest that were reported by each of the included studies. Abbreviations: DII = dietary inflammatory index; GWG = gestational weight gain; IVF = in-vitro fertilization; LAGB = laparoscopic adjustable gastric bypass; PC = prospective cohort; PCOS = polycystic ovarian syndrome; RC = retrospective cohort; RYGB = roux-en-Y gastric bypass; SES = socio-economic status; LSG = laparoscopic sleeve gastrectomy; LGA = large gestational age; AGT = abnormal glucose tolerance. ‘*’ denotes studies involving both obese and non-obese women.

## Data Availability

Not applicable.

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
