# Peer review of "Predictors of Adverse Pregnancy Outcomes in Pregnant Women Living with Obesity: A Systematic Review"

_ijerph, 2022, doi:10.3390/ijerph19042063_

Round 1

Reviewer 1 Report

Comments on the Manuscript ID: ijerph-1546324

« Predictors of adverse pregnancy outcomes in pregnant women living with obesity: a systematic review””

The paper by Fakhraei R et al. investigates by a systematic review witch women living with obesity are at higher risk of specific pregnancy complications.

This is an important question, never addressed previously, to help to determine early in pregnancy for women with obesity the care providers and the location for the maternity care. Indeed, the risks for pregnancy complications may vary widely among this population.

This article raises a relevant and useful question in practice for the management of patients, but needs some revisions.

Materials and methods section:

The paragraph 2.1 appears twice.

Results:

The main criticism is about the figures. Most of them are too difficult to read because there are too many results reported. Some of the results could be deleted for an easier reading (ex the non-significant data or those for whom different BMI categories don’t change the tendency of the results). On the other hand, not all significant results reported in the text appear in the figures.

In some figures significant results are not in bold (ex fig 4b). Some abbreviations need to be explained in the comment of the figure (ex:  fig 4a, AI/AN).

In general, data reported never point out the effect of the class of obesity on the outcome while it is shown in the figures.

Risk factors for LBW

Lines 281-282: are you sure that the studies cited reported on early weight gain? In the figure 3b, it is not specified for the Roussel 2019 study whether it is early weight gain that is considered.

Risk factors for preterm birth:

This part is very confusing with a lot of mistakes in the numbers of ref and ref cited. It should be carefully checked. For example:

Line 294: ref 47 (Dolin) is not one of the four studies reported in the figure that is probably ref 32 (Aliyu). In this last study, there was a significant difference according to maternal age.

Line 297: the remaining three studies and change the ref 47 for 32

Line 297 (end): you say there are four studies, but you cite 5 ref. In fact, there are six studies reported in this part (including number of references or data reported) and data from ref 27 (Halloran) are not reported.

Risk factors for GDM:

Line 321: ref 64 should be added within those cited because it is reported below (line 330).

Discussion:

You should add a part to discuss the impact of class of obesity: do you consider that the class of BMI changes the risk for the patient and then the decisions made early in pregnancy?

You could also have a short discussion on the relation between PE/LBW and prematurity. It is well-known that PE is associated with LBW and preterm birth. This highlights one of the limitations of your work regarding prematurity as you do not distinguish between induced and spontaneous prematurity.

Line 397: the word “were” is missing

Author Response

REVIEWER 1 – COMMENTS & RESPONSES:

Comment 1: The paper by Fakhraei R et al. investigates by a systematic review witch women living with obesity are at higher risk of specific pregnancy complications.

This is an important question, never addressed previously, to help to determine early in pregnancy for women with obesity the care providers and the location for the maternity care. Indeed, the risks for pregnancy complications may vary widely among this population.

This article raises a relevant and useful question in practice for the management of patients, but needs some revisions.

Response 1: Thank you for your comment. We have addressed each of the comments below.

Comment 2: Materials and methods section: The paragraph 2.1 appears twice.

Response 2: Thank you for this comment, we have deleted the duplicate section.

Comment 3: Results: The main criticism is about the figures. Most of them are too difficult to read because there are too many results reported. Some of the results could be deleted for an easier reading (ex the non-significant data or those for whom different BMI categories don’t change the tendency of the results). On the other hand, not all significant results reported in the text appear in the figures. In some figures significant results are not in bold (ex fig 4b). Some abbreviations need to be explained in the comment of the figure (ex:  fig 4a, AI/AN). In general, data reported never point out the effect of the class of obesity on the outcome while it is shown in the figures.

Response 3: Thank you for your comment. We have updated all our figures to address this comment but first only including significant results. Given the large number of significant data points, we were unable to discuss them all individually in the manuscript. Instead, we used our judgment to discuss the ones that are the most clinically relevant as examples throughout the text. We have reviewed all figures to ensure that the formatting is consistent and made changes as necessary. Abbreviations have been explained in the caption of the figures as well. We hope that these changes help improve the overall clarity and readability of the figures.

Comment 4: Risk factors for LBW- Lines 281-282: are you sure that the studies cited reported on early weight gain? In the figure 3b, it is not specified for the Roussel 2019 study whether it is early weight gain that is considered.

Response 4: Thank you for your comment. We have reviewed the citations, even though they do mention early weight gain in their studies, the citations are not limited to early weight gain only, so we’ve taken out the word ‘early’ to broaden the concept to include weight gain throughout the pregnancy.

Comment 5: Risk factors for preterm birth:

This part is very confusing with a lot of mistakes in the numbers of ref and ref cited. It should be carefully checked. For example:

Line 294: ref 47 (Dolin) is not one of the four studies reported in the figure that is probably ref 32 (Aliyu). In this last study, there was a significant difference according to maternal age.

Line 297: the remaining three studies and change the ref 47 for 32

Line 297 (end): you say there are four studies, but you cite 5 ref. In fact, there are six studies reported in this part (including number of references or data reported) and data from ref 27 (Halloran) are not reported.

Response 5: Thank you for your comment. This section has been thoroughly reviewed to ensure that the appropriate articles are properly referenced.

Comment 6: Risk factors for GDM: Line 321: ref 64 should be added within those cited because it is reported below (line 330).

Response 6: Thank you for your comment. The references in this section were reviewed and reference 64 was included within that group.

Comment 7: Discussion: You should add a part to discuss the impact of class of obesity: do you consider that the class of BMI changes the risk for the patient and then the decisions made early in pregnancy?

Response 7: Thank you for this comment.  We agree that this point is worth emphasizing.  A paragraph was added to the results and another paragraph was included in the discussion. 

Comment 8: You could also have a short discussion on the relation between PE/LBW and prematurity. It is well-known that PE is associated with LBW and preterm birth. This highlights one of the limitations of your work regarding prematurity as you do not distinguish between induced and spontaneous prematurity.

Response 8: We agree with this comment and have added a sentence to the limitations section to address it.

Comment 9: Line 397: the word “were” is missing

Response 9: Thank you for this comment. The version of the manuscript that we were able to download from the online platform does not have line numbers. We attempted to add line numbers to address this comment, but for some reason line numbers were not appearing even after enabling them in Word. For this reason, we were unable to locate which sentence this comment was referring to.

Reviewer 2 Report

Very interesting topic since obesity is an endemic condition nowadays.

I would appreciate a paragraph in discussion about previous bariatric surgery as a condition associated with poor pregnancy outcome.

Another issue that could be underlined is the inflammatory status associated to obesity and the pregnancy outcome

I don't fully agree about LGA (I suppose is the same as SGA) beeind detected and treated , there are screening tools and follow up means for preventing SGA associated poor outcomes

Obesity is an factor that affects also the accuracy of ultrasound scans and impact the results of the prenatal biochemical screening and also of maternal cFDNA diagnosis so mabe you could explain why you did not include those issues in the outcome and if they can be a interesting following studies.

Also I look forward to see the clinical triage tool (calculator) that would  individualize risk of pregnancy complications.

Author Response

REVIEWER 2 – COMMENTS & RESPONSES:

Comment 1: Very interesting topic since obesity is an endemic condition nowadays.

Response 1: Thank you for this comment, we agree that this is an extremely relevant topic!

Comment 2: I would appreciate a paragraph in discussion about previous bariatric surgery as a condition associated with poor pregnancy outcome.

Response 2: The importance of previous bariatric surgery as a contributor to adverse pregnancy outcomes is recognized.  Due to word limit constraints, the existing text in the discussion was expanded upon to include more details about this important, rather than adding a new paragraph entirely.

Comment 3: Another issue that could be underlined is the inflammatory status associated to obesity and the pregnancy outcome

Response 3: Thank you for this comment. We have added a paragraph in the discussion to discuss the importance of inflammatory status:

“The means by which maternal obesity impacts health of mother and child (beginning in utero and extending throughout life) are being sought.  There is increasing evidence to implicate obesity-associated low level inflammatory mediators as important contributors.  This model links maternal obesity and/or an obesogenic diet with altered adipokine secretion, insulin resistance and increased circulating lipids, which in turn is associated with increased levels of markers of inflammation (including interleukin-, interleukin-1β and tumor necrosis factor-α). Elevated levels of these factors have been associated with placental inflammation, altered placental nutrient transport and altered placental structure, all of which have the potential to negatively affect developmental programming of fetal metabolism and increase lifelong risks of obesity and metabolic syndrome.”

Comment 4: I don't fully agree about LGA (I suppose is the same as SGA) beeind detected and treated , there are screening tools and follow up means for preventing SGA associated poor outcomes

Response 4: Thank you for the comment - we agree.  There are recognized tools and guidelines for preventing SGA-associated poor outcomes.  To date, there is no similar clinical guidance for LGA pregnancies.

Comment 5: Obesity is an factor that affects also the accuracy of ultrasound scans and impact the results of the prenatal biochemical screening and also of maternal cFDNA diagnosis so mabe you could explain why you did not include those issues in the outcome and if they can be a interesting following studies.

Response 5: These aspects were discussed in the section of the manuscript related to future research endeavors.  We also support studies including these factors for predicting adverse pregnancy outcomes.

Comment 6: Also I look forward to see the clinical triage tool (calculator) that would  individualize risk of pregnancy complications.

Response 6: We agree with this comment and hope that this review can be a starting point in eventually designing a clinical triage tool.

Reviewer 3 Report

Please write in the same way: MEDLINE and Embase ... EMBASE and Medline.

In Materials and Methods - the following text is written twice: A search strategy was developed by an experienced information specialist (BS) with input from the research team. The search was conducted on January, 2017, and updated on March 5th, 2020. EMBASE and Medline electronic databases were searched. A combi-  nation of keywords and text terms were used. Literature searches were peer reviewed by a second independent information specialist using the established PRESS framework.[11] The search strategies used are provided in the Supplement to this review.

I think that the following references are incomplete:

Obesitiy in Preconception and Pregnancy. In.; 2013

Development OfEC-Oa: Obesity Update 2017. In.; 2017

Author Response

REVIEWER 3 – COMMENTS & RESPONSES:

Comment 1: Please write in the same way: MEDLINE and Embase ... EMBASE and Medline.

Response 1: The paper was revised to ensure that consistent formatting with Embase and MEDLINE were present throughout its entirety.

Comment 2: In Materials and Methods - the following text is written twice: A search strategy was developed by an experienced information specialist (BS) with input from the research team. The search was conducted on January, 2017, and updated on March 5th, 2020. EMBASE and Medline electronic databases were searched. A combi-  nation of keywords and text terms were used. Literature searches were peer reviewed by a second independent information specialist using the established PRESS framework.[11] The search strategies used are provided in the Supplement to this review.

Response 2: Thank you for indicating this. The duplicate paragraph has deleted.

Comment 3: I think that the following references are incomplete:

  • Obesitiy in Preconception and Pregnancy. In.; 2013
  • Development OfEC-Oa: Obesity Update 2017. In.; 2017

Response 3: Thank you for noticing this error – We have updated the references so that they are now complete.

Reviewer 4 Report

An interesting article, well organized, and little explored in terms of developing a clinical triage tool.

Some considerations to be discussed with the authors:

  1. The PRISMA instrument should be described according to 2020 recommendations 1016/j.ijsu.2021.105918, 10.1016/j.ijsu.2021.105906
  2. It is very interesting to use an idea regarding the differentiation of maternal-newborn predictors
  3. You mention that the average year of publication of the study was 2016 - see line 189. I do not see the point of calculating this average period, given that in Table 1 most studies were published in 2019. It would have been more attractive to graph the dynamics of the studies over time to highlight the period in which this topic was of utmost importance.

Minor comments:

  • Section 2.1 Electronic Literature Search appears twice in the text- see lines 97-102 and 104-109

Author Response

REVIEWER 4 – COMMENTS & RESPONSES:

Comment 1: An interesting article, well organized, and little explored in terms of developing a clinical triage tool.

Response 1: Thank you so much for this comment!

Comment 2: The PRISMA instrument should be described according to 2020 recommendations 1016/j.ijsu.2021.105918, 10.1016/j.ijsu.2021.105906

Response 2: Thank you for your comment. We have updated this section to indicate that the study was conducted according to the newer 2020 recommendations.

Comment 3: It is very interesting to use an idea regarding the differentiation of maternal-newborn predictors.

Response 3: Thank you for this comment. We agree that this topic is interesting and important to eventually inform the creation of a clinical triage tool.

Comment 4: You mention that the average year of publication of the study was 2016 - see line 189. I do not see the point of calculating this average period, given that in Table 1 most studies were published in 2019. It would have been more attractive to graph the dynamics of the studies over time to highlight the period in which this topic was of utmost importance.

Response 4: Thank you for this comment. We agree with the authors that a graph would be more suited for this. We have added Figure 2 which shows the number of articles published over time so that readers can see that the number was highest in 2019. We have also included this in our results. All other figures that were previously included have been renamed as a result of the addition of figure 2.

Comment 5: Section 2.1 Electronic Literature Search appears twice in the text- see lines 97-102 and 104-109

Response 5: Thank you for noticing this! We have removed the duplicate text.

Reviewer 5 Report

It is a coherent and comprehensive study. However, there are questions regarding this article:

  1. In the study eligibility criteria, it was written that there were no exclusion criteria for the included potential exposures. However, the next sentences mentioned that the outcomes of large for gestational age (LGA) newborns and macrosomia were excluded from the analysis. Why was this criterion not defined as exclusion criteria?
  2. In the result, what is the relationship between funding with this study?
  3. In study characteristics, it was written that all studies involved populations of women living with obesity, although not all studies were conducted exclusively in that population. This study aimed to investigate which women living with obesity are at higher risk of specific pregnancy complications. Why do the researcher use article that is not specific to women with the obesity population?
  4. What kind of method to analyze 61 articles were used in this article?
  5. In figure 1, what is the definition of no outcomes of interest? Does it mean that if the result of articles does not interest researchers, the article was excluded?
  6. For technical writing, the title in lines 265 and 314 may be put on the next page to read easily.
  7. It is better to put pictures with higher resolution for all figures.

Author Response

REVIEWER 5 – COMMENTS & RESPONSES:

Comment 1: In the study eligibility criteria, it was written that there were no exclusion criteria for the included potential exposures. However, the next sentences mentioned that the outcomes of large for gestational age (LGA) newborns and macrosomia were excluded from the analysis. Why was this criterion not defined as exclusion criteria?

Response 1: Thank you for this comment – We had a priori list of outcomes of interest that were identified by the study team for their clinical relevance. Thus, outcomes that were not included in this list of predefined outcomes (e.g. LGA and macrosomia) were excluded from the study. On the other hand, we wanted to be inclusive with respect to predictors (exposure) and all potential predictors that could affect the outcomes we had predefined. Therefore, we had exclusion criteria for certain outcomes based on clinical relevance, but none for the predictors themselves. The paragraph relating to the study eligibility criteria was rewritten to make this point clearer.

Comment 2: In the result, what is the relationship between funding with this study?

Response 2: Thank you for this comment. After discussing with the co-authors, we do not think that reporting of funding in the articles is necessary to mention in the results. If most of the studies were experimental (e.g. industry supported clinical trials) then reporting funding would be more important. We have therefore removed this sentence from the results.

Comment 3: In study characteristics, it was written that all studies involved populations of women living with obesity, although not all studies were conducted exclusively in that population. This study aimed to investigate which women living with obesity are at higher risk of specific pregnancy complications. Why do the researcher use article that is not specific to women with the obesity population?

Response 3: Thank you for this comment – we have added a sentence to the study eligibility criteria to explain that for studies that were not exclusively conducted among women living with obesity, only relevant data on women living with obesity were extracted in this review. Including this data allowed us to expand our sample size and capture more data on our topic.

Comment 4: What kind of method to analyze 61 articles were used in this article?

Response 4: We are not certain what the reviewers meant by this comment regarding method of analysis for the articles. If they are referring to quality assessment of the articles, then we will note that we did not implement a specific quality assessment tool to appraise the articles included. This is due to budget and time constraints as a result of the pandemic. We were forced to update the search as the systematic review took longer than anticipated to publish with delays resulting from the ongoing pandemic – this did not leave time or adequate funding for an addition of a quality assessment. The articles were reviewed multiple times by several of the co-authors as this review has gone through extensive peer review and multiple iterations with updated searches. As acknowledged in the discussion, this review is limited to observational cohort studies that are most often more prone to bias than experimental studies as a result of their study design.

If the reviewer is referring to the data extraction process, then we believe that this is well described in the methodology section of the manuscript.

Comment 5: In figure 1, what is the definition of no outcomes of interest? Does it mean that if the result of articles does not interest researchers, the article was excluded?

Response 5: As explained in the study edibility criteria, we had pre-defined a list of outcomes that were of interest for this review. Thus, “No outcome of interest” in Figure 1 refers to the instance where articles reported outcomes that were not included in our eligibility criteria reported in the methods. Please see excerpt from methodology below:

“Specific outcomes of interest included maternal mortality, stillbirths, preeclampsia, GDM, and low birth weight (LBW) including incidence of small for gestational age (SGA) newborns. The outcomes of large for gestational age (LGA) newborns and macrosomia were excluded from the analysis.”

Comment 6: For technical writing, the title in lines 265 and 314 may be put on the next page to read easily.

Response 6: We agree with this comment, however we feel that these sorts of edits will be resolved during the editorial phase of publishing.

Comment 7: It is better to put pictures with higher resolution for all figures.

Response 7: We have updated the figures to a higher resolution in the manuscript.

Round 2

Reviewer 5 Report

Thank you for the response. After we reread the article, we have some information to clarify.

  1. Figure 1, on the left side, the box for the “screening” word needs a bigger square.
  2. Table 1, what is the meaning of ** in Hedderson's (2012) article? In the footnote, we only see information for *.
  3. Please put all acronyms in figure 3a until 6b appropriately. For example, in figure 3a, there is no information about T1DM, but there is an acronym for TIDM. Moreover, it does not provide an acronym for PE.
  4. Line number 322, do you want to write “seven (7)”?

Author Response

Comment 1: Figure 1, on the left side, the box for the “screening” word needs a bigger square.

Response 1: Thank you for this comment – The figure has been updated accordingly.

Comment 2: Table 1, what is the meaning of ** in Hedderson's (2012) article? In the footnote, we only see information for *.

Response 2: Thank you for noticing this. This was an error – there should have just been a single * not double. We have updated it in the manuscript document.

Comment 3: Please put all acronyms in figure 3a until 6b appropriately. For example, in figure 3a, there is no information about T1DM, but there is an acronym for TIDM. Moreover, it does not provide an acronym for PE.

Response 3: Thank you for this comment – We have gone through all the acronyms and ensured that they correspond with the table contents. They have been updated accordingly.

Comment 4: Line number 322, do you want to write “seven (7)”?

Response 4:  Thank you – yes this was a typing error and it has been corrected.